# Metagenomic Evaluation of Bacterial and Fungal Assemblages Enriched within Diffusion Chambers and Microbial Traps Containing Uraniferous Soils

**DOI:** 10.3390/microorganisms7090324

**Published:** 2019-09-06

**Authors:** Rajneesh Jaswal, Ashish Pathak, Ashvini Chauhan

**Affiliations:** School of the Environment, 1515 S. MLK Blvd., Suite 305B, Building FSHSRC, Florida A&M University, Tallahassee, FL 32307, USA

**Keywords:** uranium, metagenomics, diffusion chamber (DC), microbial trap (MT)

## Abstract

Despite significant technological advancements in the field of microbial ecology, cultivation and subsequent isolation of the vast majority of environmental microorganisms continues to pose challenges. Isolation of the environmental microbiomes is prerequisite to better understand a myriad of ecosystem services they provide, such as bioremediation of contaminants. Towards this end, in this culturomics study, we evaluated the colonization of soil bacterial and fungal communities within diffusion chambers (DC) and microbial traps (MT) established using uraniferous soils collected from a historically contaminated soil from Aiken, USA. Microbial assemblages were compared between the DC and MT relative to the native soils using amplicon based metagenomic and bioinformatic analysis. The overall rationale of this study is that DC and MT growth chambers provide the optimum conditions under which desired microbiota, identified in a previous study to serve as the “core” microbiomes, will proliferate, leading to their successful isolation. Specifically, the core microbiomes consisted of assemblages of bacteria (*Burkholderia* spp.) and fungi (*Penicillium* spp.), respectively. The findings from this study further supported previous data such that the abundance and diversity of the desired “core” microbiomes significantly increased as a function of enrichments over three consecutive generations of DC and MT, respectively. Metagenomic analysis of the DC/MT generations also revealed that enrichment and stable populations of the desired “core” bacterial and fungal microbiomes develop within the first 20 days of incubation and the practice of subsequent transfers for second and third generations, as is standard in previous studies, may be unnecessary. As a cost and time cutting measure, this study recommends running the DC/MT chambers for only a 20-day time period, as opposed to previous studies, which were run for months. In summation, it was concluded that, using the diffusion chamber-based enrichment techniques, growth of desired microbiota possessing environmentally relevant functions can be achieved in a much shorter time frame than has been previously shown.

## 1. Introduction

Despite the tremendous progress in microbial ecology techniques, cultivation of the soil microbiomes still remains challenging and, consequently, less than 1% of the total soil microbiome is amenable to cultivation under standard laboratory conditions [1]. A variety of different reasons have been assigned to the low success in the cultivation of environmental microbiomes, including, but not limited to, soil dilution prior to plating (eliminates bacteria that exist in extremely low numbers), media components (typically benefitting faster growing bacteria), and media devoid of cofactors or cell-to-cell signaling molecules, usually produced by bacterial communities in their native habitat, that facilitate microbial growth [2]. To improve microbial growth and isolation from environmental samples, several approaches have recently been proposed, such as varying media composition and growth conditions [3], high throughput extinction culturing [4], high throughput single-cell encapsulation [5], diffusion chambers [6], using different gelling agents, antioxidants, or signaling molecules [7,8], and even increasing incubation times along with reducing nutrient concentrations [9]. Other researchers have shown that simple alterations in the media preparation, such as autoclaving the phosphate source and agar separately from other media components to reduce the oxidative stress, can improve the cultivability of microbes [10]. Nguyen et al., proposed using growth media prepared from soil extract using an 80% methanol water mixture to facilitate the extraction of organic and volatile compounds [11].

In ongoing studies, our laboratory is heavily engaged on the application of metagenomics to survey the soil microbiome such that a better understanding on the bacterial and fungal diversity, along with their environmentally relevant functions, can be obtained [12]. Specifically, when metagenomics was coupled to study the enrichment of soil-borne microbiota in diffusion chambers (DC), this approach emerged as a holistic assessment of the bacterial and fungal assemblages colonizing the chambers. Overall, this can be a precise and sensitive tool to assess the enrichment process and even dictate the optimum incubation time for isolation of specific microbiota with the desired functions. For example, in a recent study, we showed the microbial diversity in DC and MT (Microbial Trap) experiments to be dominated by Proteobacterial phyla and *Burkholderia* genus [12]. Notably, most microbial ecology studies focus on bacterial communities with little to no emphasis given on the soil-borne fungi, despite their propensity to outcompete bacteria at higher concentrations of environmental contaminants, such as U [12], and other contaminants [13,14,15]. Towards this end, as part of a larger ongoing study, we are focused on the bioremediative roles of not only bacteria, but also fungal communities, within the heavy-metal contaminated Savannah River Site, SC (SRS) soils [16,17,18]. Specifically, in our previous study on the evaluation of the DC and MT techniques to enrich and isolate U-resistant fungal microbiota from the uranium-rich SRS soils, a predominance of Ascomycota phylum, with *Penicillium* as the most dominant genus, was demonstrated [18]. Interestingly, these dominant bacterial and fungal strains were then isolated and found to possess environmentally beneficial traits of U bioremediation. In line with our above referenced study [18], application of metagenomics to SRS soils facilitated identification of the ecological diversity, with functions related to the detoxification of contaminants [16,17], such as the presence of a suite of genomic-determinants for resistance, membrane transport, and efflux, as also shown for other metal contaminated ecosystems [19,20]. Thus, with metagenomics combined with DC and MT, potentially novel U-remediative bacteria can be identified and isolated, as shown in our recent report [6]. 

Note that the diffusion chambers are diluted environmental samples that are embedded within sterile agarose plugs and are sealed by semi-permeable membranes on both sides and placed on moist soils. This facilitates crossover of soil-borne nutrients and other molecules as well as permitting microbial interactions to occur for the enrichment of environmental microbiota [6,21]. Not only bacteria, but potentially novel soil-borne fungi are also encouraged to grow within the MT chambers, as was shown in our recent study, which led to the isolation of a novel *Penicillium* species, as revealed by whole genome sequencing, annotation, and bioinformatic analysis [12]. However, it is standard practice to incubate the DC/MT chambers on moist soil samples for at least 30 days (generation 1), followed by collection of the biomass from gen1 and transfer into freshly prepared chambers, called gen2, with further incubation of 30 days before transferring into the third gen. After incubation of each gen, biomass is screened for desirable microorganisms, as shown previously [6,12]. This way, at least 3 transfers over a 60-day period are typically required prior to the successful isolation of environmentally relevant bacteria [6,12,21]. The central question we asked in this study is “how many generations does it take to reach stable populations of the desired bacterial and fungal “core” microbiomes relevant to desired trait(s) under investigation?” Furthermore, the changes within the soil microbial community composition over the course of DC/MT incubation times are not fully understood.

To address the above stated questions, DC and MT chambers were established using soil samples collected from the Savanah River Site (SRS) located near Aiken, South Carolina. SRS has historically been exposed to large quantities of U released from decades of nuclear material production activities and the refinement of nuclear materials [22]. Because many groups of bacteria and fungi are known to be resistant to and bioremediate U, DC and MT chambers result in the “in-situ” proliferation of a diverse assemblage of the soil microbiome, facilitating isolation and evaluation of bioremediative microbiomes [6,18,23]. To further advance this research, we used a slightly modified, high-throughput, version of the DC and MT and followed the development of both bacterial and fungal assemblages over gen 1–3 and coupled this with metagenomics, which suggested that the desired “core” bacterial and fungal microbiomes developed stably within the first 20 days of incubation and the practice of subsequent transfers for second and third generations, as is standard in previous studies, may not be necessary. As a cost and time cutting measure, this study recommends running the DC/MT chambers for only a 20-day time period, as opposed to previous studies that were run for months.

## 2. Material and Methods

### 2.1. Soil Sample Collection

Sub-surface soil samples were collected from site 101, which is an abandoned pond (Steeds Pond) that served as a natural settling basin along the Tims Branch stream corridor on the SRS, located near Aiken, South Carolina. Approximately 4.2 mM of total U was estimated to be present in the soil samples used in this study for the establishment of DC/MT chambers [17]. It must be noted that this U concentration is several orders in magnitude higher than that recognized by EPA as toxic [24]. Triplicate surficial soil samples were collected from site 101, stored on ice, shipped overnight to the laboratory, and immediately processed for establishment of diffusion chambers (DC) and microbial traps (MT). All experiments were initiated within a day of sample collection, thus minimizing artifacts resulting from delayed sample processing for downstream experiments. Chemicals used in this study were analytical grade and purchased from VWR (Atlanta, GA, USA), unless otherwise mentioned.

### 2.2. Establishment of Diffusion Chambers/Microbial Traps (DC/MT)

To establish the DC/MT, we used a plate with several even sized holes, 0.8 mm in height and 0.7 mm in diameter. Nine of such holes were used together as multiple chambers for a single DC or MT microbiome establishment (Figure 1). Specifically, a 0.03 µm pore size polycarbonate membrane (GE Healthcare Biosciences, Pittsburgh, USA) was glued to one side of the tray (using silicone glue) in such a way that the nine holes on the tray would fall exactly towards the center of the membrane, to increase throughput microbial cultivation. For establishing DC, a 1 g soil sample was mixed with 9 mL of sterile normal saline and serially diluted to 10^−3^ dilution. One ml of this diluted sample was mixed with 9 mL of sterile molten agar at 45 °C to achieve a final concentration of 10^−4^. Approximately 435 µL of the 10^−4^ diluted soil sample in agar was added to each of the nine wells to completely fill the holes. After the agar solidified, a second 0.03 µm pore size membrane was glued on top to seal the chamber, thus establishing DC. The premise of this approach is that the 0.03 µm pore-size membrane permits the diffusion of nutrients into the chamber and facilitates the growth of environmental microbiota within the agar by mimicking in-situ growth conditions [18].

For establishing MT, 435 µL of sterile molten agar at 45 °C was added to each of the nine wells (of a different plate). After the agar solidified, a 0.2 µm pore size membrane was glued on top to seal the chamber, thus establishing MT. Note that no soil sample was added to the MT and that the approach was used for the isolation of uncultivable actinomycetes, many of which are well-known for their biodegradative and metabolic potential. The premise of this approach is that the 0.2 µm pore-size membrane permits the actinomycetes mycelia through the filter pores to the underlying agar medium while restricting the entrance of the soil bacteria to the agar. The DC plate (any side facing down) and MT plate (0.2 µm pore size, side facing down) were then placed on the U contaminated soil (approximately 5 mm thick) and incubated at 28 °C for 20 days, thus starting 1^st^ generation (or Gen 1) of the DC/MT. The soil underneath the plates was mixed every two days to remove the buildup of anaerobic conditions. The DC plate were also flipped every two days while the MT plate was not flipped. After 20 days of incubation, chambers were opened and the gel-embedded biomass inside the DC and MT chambers was collected and homogenized by passaging through a 22-gauge needle syringe, twice. Part of this homogenized gel was used for DNA isolation, part was used for bacterial and fungal isolations on LB agar (Luria Bertani) and PDA (Potato Dextrose Agar) respectively, and part was used as inoculum for the Gen 2 of DC and MT, which were further incubated for 20 days at 28 °C on the U contaminated soil. Mixing of the underneath soil and flipping the of DC tray were performed just as the incubation of Gen 1. The same procedure was performed at the end of the Gen 2 incubation period, as was done at the end of Gen 1, and Gen 3 commenced in similar conditions.

### 2.3. Microbiome Analysis

Genomic DNA was isolated from soils and DC/MT plugs using the DNeasy PowerLyzer Kit, according to the manufacturer’s instructions (Qiagen Inc., Germantown, MD, USA). The quantity and quality of the isolated genomic DNA was evaluated using a micro-volume spectrophotometer (NanoDrop Technologies, Wilmington, DE, USA) and further processed for 16S metagenomics. Briefly, sequence libraries were prepared using the Illumina Nextera XT kit, according to the manufacturer’s instructions (Illumina Inc., San Diego, CA) using primers 515F/926R for bacteria [25] and ITS1F/ITS2R for fungi [26]. Sequencing was performed on an Illumina NextSeq500 instrument employing a mid-output kit with 2 × 150 paired-end sequencing. Forward and reverse reads were merged using the PEAR (Paired-End reAd merger) approach [27] and the merged reads were trimmed to remove ambiguous nucleotides and primer sequences, and trimmed based on the quality threshold of *p* = 0.01. Reads that lacked either primer sequence and any sequences less than 300 bp or 100 bp were discarded for the 515F/926R and the ITS1/ITS2 primer sets, respectively. Chimeric sequences were identified and removed using the USEARCH algorithm with a comparison to Silva v132 reference sequence database for the 515F/926R primer sets [28,29]. Meanwhile, chimeric sequences for the ITS1/ITS2 primer set were identified, and subsequently removed, using UCHIME in a de novo fashion [28].

The standard QIIME pipeline was modified to generate taxonomic summaries using sub-OTU (operational taxonomic units) resolution of the sequence dataset [30,31]. Briefly, the resulting sequence files were then merged with sample information. All sequences were then dereplicated to produce a list of unique sequences. All sequences that had an abundance of at least 10 counts were designated seed sequences. USEARCH was then used to find the nearest seed sequence for any non-seed sequence with a minimum identity threshold of 98%. For any non-seed sequence that matched a seed sequence, its counts were merged with the seed sequence counts. For any non-seed sequence that did not match a seed sequence it would remain an independent sequence. 

Taxonomic annotations for seed and unmatched non-seed sequences were assigned using the USEARCH and Silva v132reference with a minimum similarity threshold of 90% for the 515F/926R primer set [28,29] and the UNITE ITS reference was used for the ITS1/ITS2 primer set [32]. In order to improve the depth of annotation, the standard QIIME assignment algorithm was modified to only consider hits at each taxonomic level that had an assigned name. For example, a reference annotated as “Bacteria; Firmicutes; Clostridia; Clostridiales; Ruminococcaceae;” would be considered in the assignment of the taxonomic kingdom through family, but would not be used for the assignment of the genus or species. Furthermore, we utilized the criteria that any hits in the reference database must have a minimum identity of 97% or 99% to be considered for genus or species level assignment, respectively. Taxonomic annotations and sequence abundance data were then merged into a single sequence table separately for each primer set. Microbiome Analyst [33] was run on the processed sequence data, as stated above, to identify the “core” microbiome and perform statistical and diversity analysis. Using the microbiome analyst workflow, sequence data were filtered for low count and low variance using default parameters, which resulted in the removal of 227 low abundance features based on prevalence and 40 low variance features based on inter-quantile range (iqr). A total of 610,718 bacterial sequence read counts were inputted into the workflow, which resulted in the binning of 1717 operational taxonomic units (OTUs). Similarly, 114 fungal OTUs were detected from a total of 119,204 fungal sequence read counts with removal of 10 low abundance features based on prevalence and 3 features based on iqr, respectively. 

Libraries were then rarefied to the minimum library size and total sum scaling was performed to account for sample variability such that biologically meaningful comparisons could be drawn. The “core” microbiome refers to the set of genus level taxa that were detected in a high fraction across the tested soils using the following threshold levels: Sample prevalence (20%) and relative abundance of 0.2%. Dendrogram analysis was run at the genus level using the Bray–Curtis index and selecting the experimental factor as grouped in the metadata file. Further ordination analysis on the amplicon-based metagenomics data was performed at the genus level using the built-in MicrobiomeAnalyst tools, such as α-diversity (Chao1 measure with T-test/ANOVA), β-diversity plotted as PCoA (Bray–Curtis distance method with PERMANOVA), univariate analysis using T-test/ANOVA with an adjusted cutoff value of 0.05, and differential abundance analysis at the genus level, calculated using EdgeR at an adjusted *p*-value cut off of 0.05.

### 2.4. Metagenomic Sequence Accession Number

The metagenomic sequences obtained from this study are available from NCBI’s Sequence Read Archive/European Nucleotide Archive, accession numbers SAMN12136482, SAMN12136483, SAMN12136485, SAMN12136486, SAMN12136498, SAMN12136499, SAMN12136511, SAMN12136512, SAMN12136524, SAMN12136525, SAMN12136537, SAMN12136539, SAMN12136551, SAMN12136552, SAMN12136564, SAMN12136565, SAMN12136577, and SAMN12136578, under BioProject PRJNA550441.

## 3. Results and Discussion

### Changes in Bacterial and Fungal Communities Between the DC/MT Generations for Uranium Contaminated Soil

Sequence read counts for bacteria and fungi domains obtained from the metagenomic analysis of the biomass obtained from different generations of DC and MT were bioinformatically binned at the phylum and genus levels. The results are shown in Table 1. As shown in the table, the majority of the retrieved sequences were taxonomically resolved at the phylum level. 

Figure 2A shows the relative abundance of the major phyla that were identified across DC/MT generations originating from U contaminated soils as evaluated by 16S based metagenomics. Specifically, *Proteobacteria*, *Bacteroidetes*, *Verrucomicrobia*, *Acidobacteria*, and *Actinobacteria* were identified as the most prominent phyla in the U soil DC chambers. *Proteobacteria* and *Actinobacteria* have previously been reported in the U contaminated soils in southern China [34]. Moreover, our previous observations in the U contaminated SRS soil also mirrors these findings, such that *Proteobacteria*, *Acidobacteria*, and *Actinobacteria* were identified as the dominant phyla [18]. Similar dominance of the *Proteobacteria* and *Acidobacteria* community has been shown in the U contaminated soils of Limousin, France [35]. It is likely that the rationale for *Proteobacteria* dominance is related to the ability of many proteobacterial members to resist and bioremediate heavy metals, hence the SRS uraniferous habitat also likely selected for these lineages. Note that *Proteobacteria* was the dominant phyla in DC gen1, with approximately 80% relative abundance in comparison to all the other phyla. However, in the subsequent generations, their overall abundance reduced to approximately 70% and 45% in the gen2 and gen3, respectively. On the other hand, the relative abundance of *Bacteroidetes* and *Verrucomicrobia* increased on sub-culturing to gen2 and gen3. Rarefaction curves drawn for each generation of DC and MT showed that the bacterial species diversity was almost similar, between 300–350 (Figure 3), indicating that the DC treatment was not detrimental to the proliferation of bacterial diversity.

At the genus level, *Burkholderia*, *Rhodanobacter*, *Mucilaginibacter*, *Prosthecobacter*, *Bradyrhizobium*, and *Dyella* were identified as the most dominant bacteria over 100 different genera identified in these samples (Figure 2B). These bacterial groups, including *Rhodanobacter* [36], and *Dyella* spp. [37], have also been reported to survive and colonize uraniferous soils, which is in line with findings from this study. Furthermore, our previous studies in the U contaminated SRS soil also assert the dominance of *Burkholderia*, and *Rhodanobacter* as the dominant genera [16,18,38]. Other studies have also found that *Burkholderia*-like microorganisms comprise a significant proportion of the U reducing microbiota [39,40,41,42]. *Burkholderia* spp. ubiquitously occurs in the contaminated environment due to their rigorous biodegradative and metal resistance abilities [43,44,45,46,47], including to uranium [48]. Hence, it can be rationalized that *Burkholderia* spp. not only survives in uraniferous soils, but may also likely play a critical role in the microbially-mediated remediation of U. It must also be noted that about 23% of the genera, in the DC gen1, were not recognized in the 16S database, which is highly suggestive of novel microorganisms colonizing the uranium-rich soil habitat. More importantly, the overall abundance of these unidentified genera increased to approximately 35–38% in gen2 and gen3. These observations emphasize the fact these novel isolation techniques, such as DC, can be used to further enrich previously unculturable microbes and exploit their functional traits.

The MicrobiomeAnalyst pipeline was used to identify the “core” microbiome and analyze diversity. The “core” microbiome identifies the genus level taxa that were detected in a high fraction using the following threshold levels: Sample prevalence (50%) and relative abundance of 0.2%. At the genus level, *Burkholderia*, *Bradyrhizobium*, and *Rhodanobacter* were identified to be most abundant in the DC/MT treatments (Appendix A). When *Burkholderia*, *Bradyrhizobium*, and *Rhodanobacter* species were further evaluated using differential abundance statistical analysis, it was clear that the counts for these groups increased across the generations, i.e., incubation times in the chambers (Figure 4). The conditions presented to the soil microbiome within the DC/MT agar plugs clearly enhances growth of those communities that can persist in the soils and, by inference, likely provide ecosystem services. Interestingly, it was observed that these bacterial assemblages persisted within the MT agar plugs. Note that the MT treatments were started with sterile agar with a 0.2 µm filter membrane placed on the bottom, so the bacteria from the bottom soils are not expected to cross over into the agar chamber; only the soil-borne actinomycetes are expected to colonize the chambers as they form filamentous structures. It is likely that the fungal filaments form conduits for the transfer of bacteria or damage the membranes facilitating microbial cross-over. Furthermore, dendrogram analysis of the bacterial community in the DC/MT gels confirmed that the DC/MT treatments were differentially selective to the soil microbiome (Figure 5). The first generation of both DC and MT clustered separately from the 2^nd^ and 3^rd^ generations, respectively. The microbiome in the 2^nd^ and 3^rd^ generations of DC/MT were perhaps more similar to each other than the other treatment, as they clustered together in the dendrogram. These observations also suggest that the incubation times have a significant impact on the diversity selection and growth of the microbiome.

Since fungi are known to outcompete bacteria at high uranium concentrations [12,49], the effect of DC/MT culture conditions on fungal community structures was also evaluated as part of this study. The Ascomycota phylum was present at a very high relative abundance of 60% in the gen1 DC, which increased slightly in the subsequent generations (Figure 6A). The phyla Basidiomycota was observed as the second most abundant phylum, which increased significantly in relative abundance from DC gen1 to gen2 and then remained almost stable in gen3. On the other hand, the relative abundance of Zygomycota reduced with increasing DC generations. The fungal community structure in the MT was entirely dominated by Ascomycota phylum. MT gen1 predominantly comprised of about 50% Ascomycota and approximately 45% Zygomycota. However, in the subsequent generations, Ascomycota completely outgrew Zygomycota and the other phyla, indicating that the MT treatments were highly favorable to the growth of Ascomycota.

At the genus level, *Cryptococcus* was the most abundant in the DC gen1 and its relative abundance increased in the gen2 and remained at the same level in gen3 (Figure 6B). The genus *Trichoderma* was the second most abundant genus in the DC gen1 and increased in gen2 and gen3. The relative abundance of ‘other’ unidentified genera, present in very small numbers, were observed to reduce in the DC studies on subsequent culturing. In the MT studies, all the three generations were dominated by ‘other’ genera and the MT conditions were suitable for their relative abundances to increase in subsequent generations. MT gen1 did have a significant abundance of genus *Mortierella*, but it was not seen in subsequent generations at all. Genus *Lecythophora* was observed in MT gen1, but its numbers were seen to be reduced in the subsequent generations. Our previous DC/MT studies on the uranium soils were dominated by the genera *Penicillium* and *Aspergillus* [18], however, in the current study, the relative abundance of *Penicillium* reduced in subsequent generations for both DC and MT. *Aspergillus*, on the other hand, was almost negligible in any generation of DC/MT. It can be summed that the above stated fungal genera, in concert, likely resist and perform U remediative functions in the SRS soils.

The “core” microbiome for the fungal communities, regardless of DC and MT treatments, were identified as *Penicillium* and *Trichoderma* (Appendix A), which are similar to our previous study. When differential analysis was performed for the fungal communities, it was found that *Penicillium* spp. and *Trichoderma* spp. abundance increased from MT gen1 to gen2 and then reduced in gen 3 (Figure 7). In the DCs, the number of *Penicillium* reduced on subculturing. *Trichoderma*, on the other hand, was favored in both MTs and DCs, with their abundances increasing on subsequent subculturing.

When the different bacterial and fungal assemblages developed across the three generations were compared using differential analysis, it was clear that stable populations of the “core” microbiomes developed as early as the first gen (Figure 4A–C and Figure 7A,B). These “core” microbiomes which were also previously shown to possess U-remediative functions, mainly belonged to *Burkholderia*, *Bradyrhizobium*, and *Rhodanobacter* in the bacterial groups, along with *Penicillium* and *Trichoderma* in the fungal groups, respectively. Therefore, this study suggests that the practice of subsequent transfer of gen1 biomass for the second and third generations, as is standard in previous studies [6,12,21], is likely to be redundant and unnecessary. As a cost-cutting measure and to save time, we recommend running the DC/MT chambers for only a 20-day time period, as opposed to previous studies that were run for months.

Furthermore, statistical differences between the DC/MT generations were assessed using the α and β indices of diversity at the genus levels and ordinations obtained were plotted as PCoA using the Bray–Curtis index (Figure 8). The α diversity is a measure of the diversity within each sample, while the β diversity is a measure of the diversity between the evaluated sample set. It should be noted that α diversity in the 1^st^ generation of DC was higher relative to the 1^st^ generation culture of MT, which is obviously due to the fact that soil samples were mixed with agar in the DC, while the 1^st^ generation of MT was started sterile (without the addition of the soil microbiota). As seen from the α diversity plot (Figure 8A), the diversity in the DC microbiome reduced from gen 1 to gen 2, suggesting that growth of certain microbiota was favored by the culture conditions. However, it must be noted that this initial growth stress was overcome in the 3^rd^ generation of DC, which displayed higher biodiversity than the 2^nd^ generation, and slightly less diversity than the 1^st^ DC generation. These results indicate the resilience of the soil microbiota to overcome the initial stress of the artificial growth conditions. However, α diversity in the MT microbiome reduced in the subsequent generations, suggesting the selection of a select few genera suitable to survive in the MT growth conditions. The β diversity plot demonstrated that the DC and MT generations were quite different from one another (Figure 8B). Notably, generation 1 for both DC and MT were clustered far away from the 2^nd^ and 3^rd^ generations of the respective microbiome. Whereas the 2^nd^ and 3^rd^ generation of either DC or MT were clustered together to each other, indicating the similarities in the biodiversity of the microbiomes. It must also be noted that the gen 1 DC was located away from the gen 1 MT, showing high difference in the biodiversity. Overall, this study provides a detailed description of changes that are observed within bacterial and fungal assemblages over three generations of DC and MT chambers, resulting in a better understanding of the structural-functional relationships of environmental microbiota in uraniferous soils.

## Figures and Tables

**Figure 1 microorganisms-07-00324-f001:**
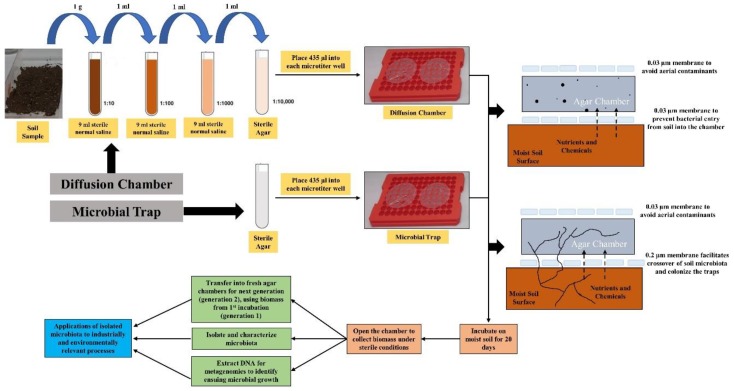
Overall schematic of experimental design to establish the DC/MT chambers.

**Figure 2 microorganisms-07-00324-f002:**
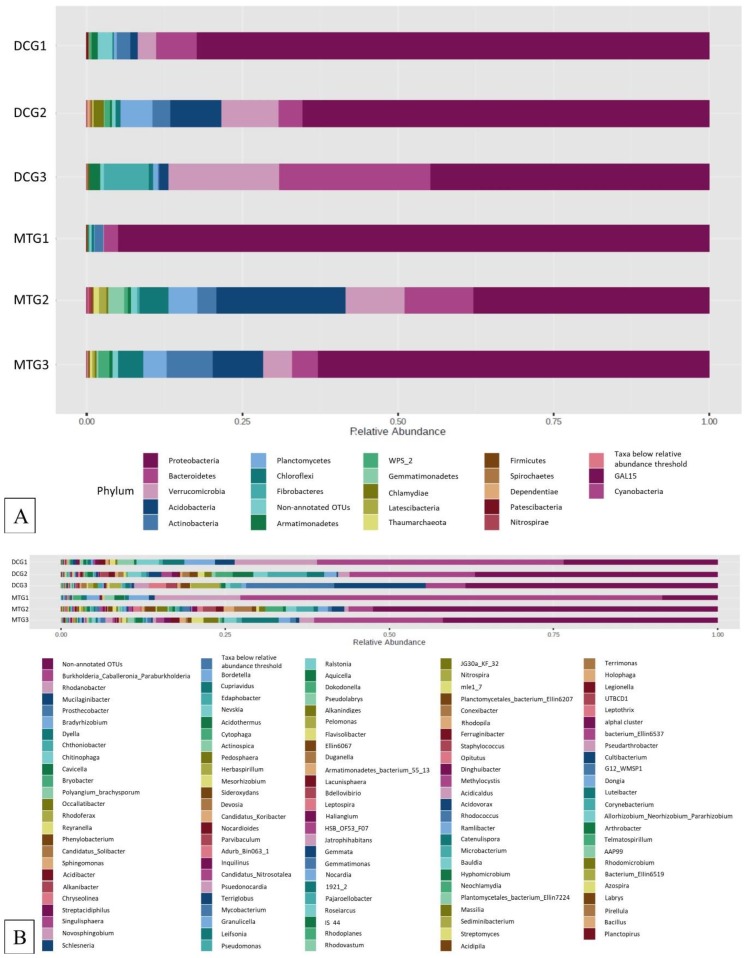
Bacterial diversity plotted as relative abundances shown at the phyla (**A**) and genus (**B**) levels, identified from the soils and the DC/MT chambers, respectively.

**Figure 3 microorganisms-07-00324-f003:**
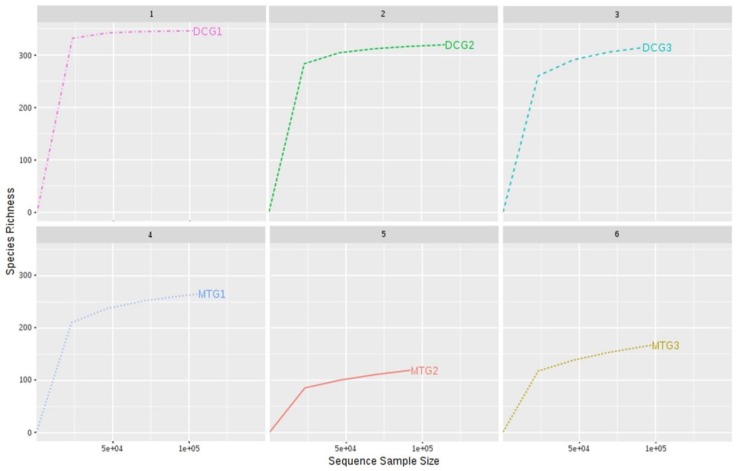
Rarefaction curves drawn for each generation of DC/MT, indicating the proliferation of biomass.

**Figure 4 microorganisms-07-00324-f004:**
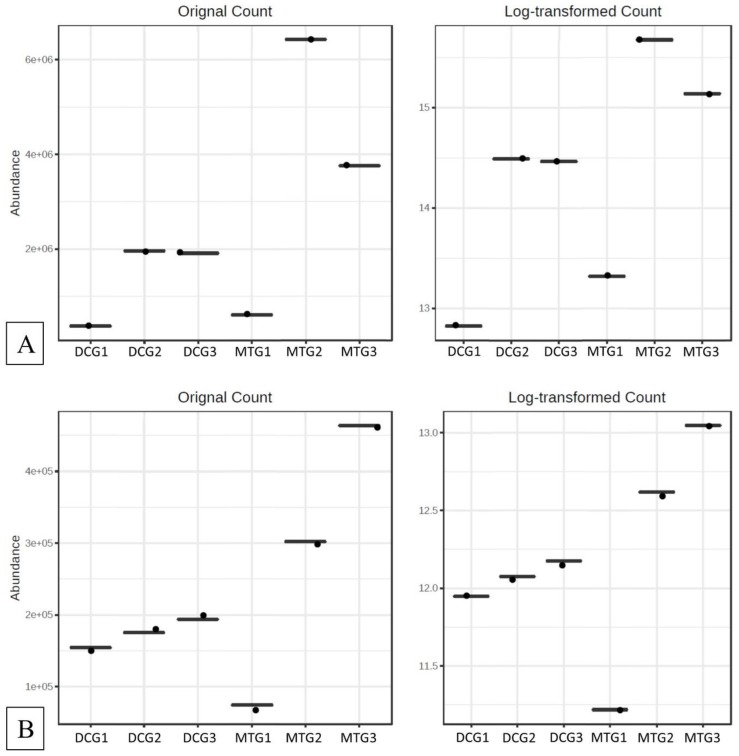
Differential abundance statistical analysis for (**A**) *Burkholderia*, (**B**) *Bradyrhizobium*, and (**C**) *Rhodanobacter* species in the DC/MT generations.

**Figure 5 microorganisms-07-00324-f005:**
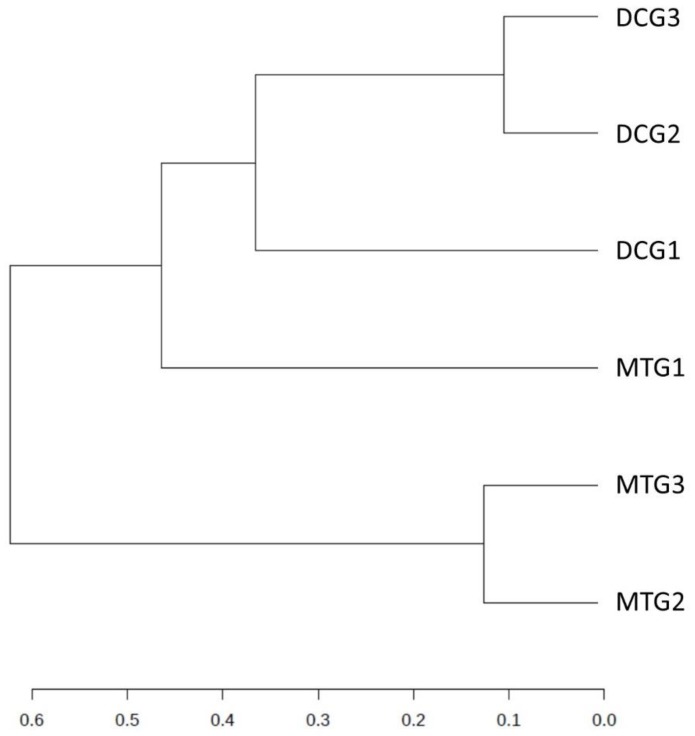
Dendrogram analysis at the DC/MT generations using the Bray–Curtis index and the ward clustering algorithm.

**Figure 6 microorganisms-07-00324-f006:**
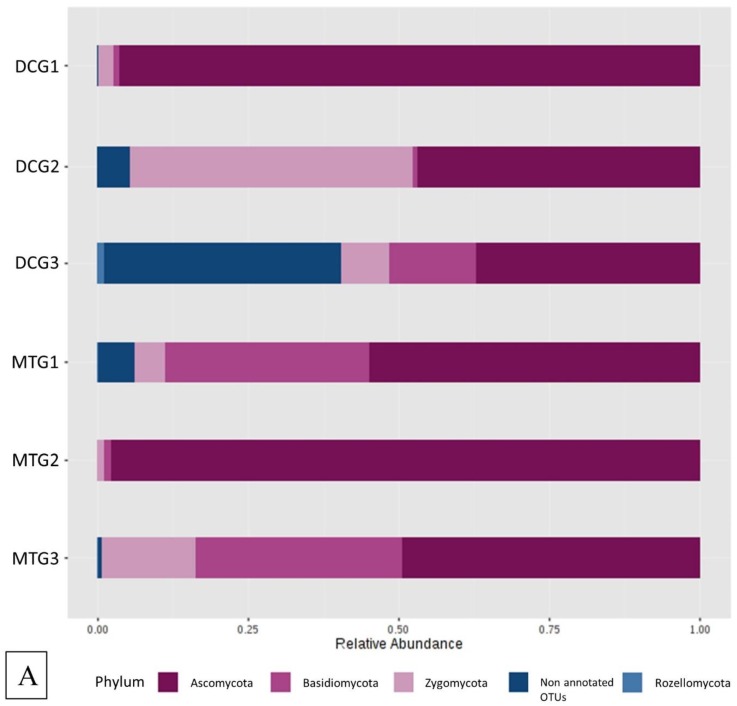
Fungal diversity plotted as relative abundances shown at the phyla (**A**) and genus (**B**) levels, identified from the soils and DC/MT chambers, respectively.

**Figure 7 microorganisms-07-00324-f007:**
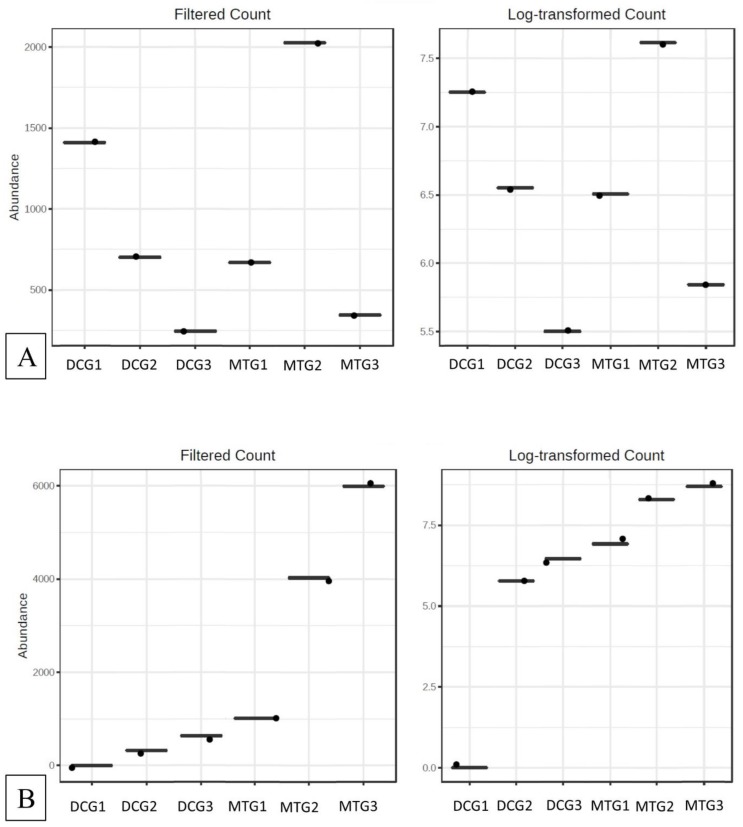
Differential abundance statistical analysis for (**A**) *Penicillium* and (**B**) *Trichoderma* species, in the DC/MT generations.

**Figure 8 microorganisms-07-00324-f008:**
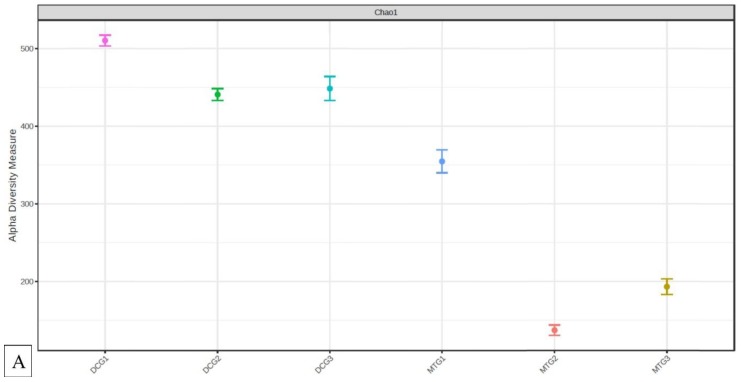
Shown are the α (**A**) and β diversity (**B**) analysis and significance testing between the soils and DC/MT chambers. A [PERMANOVA] R-squared: 0.9976; *p*-value < 0.066667 was obtained from this analysis.

**Table 1 microorganisms-07-00324-t001:** Taxonomic assignment of metagenomic sequences obtained from different generations of the diffusion chamber (DC) and microbial trap (MT) established with uranium contaminated soils.

Sample Name	Total Read Counts	Phylum Level Annotation	Genus Level Annotation
**Domain Bacteria:**			
DC-generation 1	106,514	105,499	50,468
DC-generation 2	120,145	119,179	67,456
DC-generation 3	92,696	92,121	58,014
MT- generation 1	109,144	108,476	66,457
MT- generation 2	99,139	98,791	84,591
MT- generation 3	100,637	98,342	75,028
**Domain Fungi:**			
DC-generation 1	23,564	14,870	6186
DC-generation 2	40,409	40,167	21,498
DC-generation 3	30,051	28,305	17,923
MT- generation 1	22,288	21,071	12,346
MT- generation 2	23,073	23,058	2791
MT- generation 3	27,535	27,488	1225

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
