# Peer review of "Metagenomic Evaluation of Bacterial and Fungal Assemblages Enriched within Diffusion Chambers and Microbial Traps Containing Uraniferous Soils"

_microorganisms, 2019, doi:10.3390/microorganisms7090324_

Round 1

Reviewer 1 Report

The study provides interesting information about bacteria in uriniferous soils. Unfortunately, at the present state, the manuscript is not suitable for publication and a major revision is needed. Following, my suggestions to improve the manuscript:

My main concern is about the material and method section:

-Please, explain better the experimental plan (number of replicates and samples used for the DNA extraction and sequencing). A supplementary figure explaining the experimental plan and some pictures of the diffusion chambers and of the microbial traps could be useful.

-Please, give more details on the bioinformatic pipeline used. At line 162-164, authors report the use of DIAMOND and MEGAN for the taxonomic assignment, but at line 165 they say: “MicrobiomeAnalyst pipeline [28] was run on QIIME processed sequence data…”. Which pipeline did they use for the bioinformatic analysis?

-More details on the quality filtering and the sequence identity used are needed.

-Line 165. MicrobiomeAnalyst is not a pipeline, but a it’s web-based tool for statistical analysis and data visualisation.

-In this section, they report the use of 16S metagenomics approach to investigate bacterial and fungal soil communities, showing a taxonomic assignment for both groups. I was wondering, how have they performed the fungal OTU assignment? 16S approach, amplifying a specific region of the 16S rRNA gene sequences, is specific for prokaryotes.

-Please, improve this section with details on the 16S region amplified and on the primer sequences and references used.

-As reported in the journal guidelines:

“Deposition of Sequences and of Expression Data. New sequence information must be deposited to the appropriate database prior to submission of the manuscript. Accession numbers provided by the database should be included in the submitted manuscript. Manuscripts will not be published until the accession number is provided.”

Authors must report in the manuscript the information about the data repository.  

In general Result and Discussion section is well written, but the paragraph on the fungal community description should be revised according to my previous comment. Following some comments on this section:

-Line 202-204. Rarefaction allows the calculation of species richness for a given number of individual samples, based on the construction of a rarefaction curves.  The curve shows the number of different species as a function of the number of samples. For that reason, there is no correlation between rarefaction curve (number of species) and microbial biomass. Please revise this sentence.

-Authors should report the number of annotated reads for each taxonomic level they have showed in the manuscript.

Please, revise all the manuscript for a correct use of the abbreviations. Following some examples:

Line 62: please, explain MT abbreviation

Line 68: please, explain SRS abbreviation

Line 77: DC is already explained before

Line 117: DC and MT are already defined

Please, revise all the figure legends. They should be self-explanatory.

(e.g. Figure 1, please, order properly the samples (DCG1, MTG1, DCG2, MTG2, DCG3, MTG3) and explain name in the legend.)

Author Response

Please see attached file, thanks.

Following is our response to the reviewer’s comments. Shown in black color are the reviewer’s comments and the author’s responses appear in red color.

Reviewer # 1

-Please, explain better the experimental plan (number of replicates and samples used for the DNA extraction and sequencing). A supplementary figure explaining the experimental plan and some pictures of the diffusion chambers and of the microbial traps could be useful.

Author’s response: To address the confusion around the experimental plan, we have included a newly generated figure 1 that outlines the entire protocol as a schematic workflow.

-Please, give more details on the bioinformatic pipeline used. At line 162-164, authors report the use of DIAMOND and MEGAN for the taxonomic assignment, but at line 165 they say: “MicrobiomeAnalyst pipeline [28] was run on QIIME processed sequence data…”. Which pipeline did they use for the bioinformatic analysis?

Author’s response: We have revised this section to specify the bioinformatics workflow which was not accurately stated in the previous version.

-More details on the quality filtering and the sequence identity used are needed.

Author’s response: As stated above, we have revised this section to specify the bioinformatics workflow including the data quality assessment and filtration etc., which was not accurately stated in the previous version.

-Line 165. MicrobiomeAnalyst is not a pipeline, but a it’s web-based tool for statistical analysis and data visualisation.

Author’s response: We have revised this sentence.

-In this section, they report the use of 16S metagenomics approach to investigate bacterial and fungal soil communities, showing a taxonomic assignment for both groups. I was wondering, how have they performed the fungal OTU assignment? 16S approach, amplifying a specific region of the 16S rRNA gene sequences, is specific for prokaryotes.

Author’s response: We appreciate your detailed review and for identifying this oversight. Please note that we have now revised the bioinformatics section to further clarify how bioinformatic analysis was conducted on the bacterial and fungal metagenome sequences.

-Please, improve this section with details on the 16S region amplified and on the primer sequences and references used.

Author’s response: We have revised this section to include details on primer sets used for bacterial and fungal metagenome sequencing and ensuing bioinformatics.

-As reported in the journal guidelines:

“Deposition of Sequences and of Expression Data. New sequence information must be deposited to the appropriate database prior to submission of the manuscript. Accession numbers provided by the database should be included in the submitted manuscript. Manuscripts will not be published until the accession number is provided.”

Author’s response: We have included a new section 2.4 under materials and methods, which states the short read archive (SRA) accession numbers for the bacterial and fungal metagenome sequences obtained. 

Authors must report in the manuscript the information about the data repository.

Author’s response: As stated above, a new section 2.4 under materials and methods, which states the short read archive (SRA) accession numbers for the bacterial and fungal metagenome sequences obtained.

In general Result and Discussion section is well written, but the paragraph on the fungal community description should be revised according to my previous comment. Following some comments on this section:

-Line 202-204. Rarefaction allows the calculation of species richness for a given number of individual samples, based on the construction of a rarefaction curves.  The curve shows the number of different species as a function of the number of samples. For that reason, there is no correlation between rarefaction curve (number of species) and microbial biomass. Please revise this sentence.

Author’s response: We have revised this section to be more coherent.

 -Authors should report the number of annotated reads for each taxonomic level they have showed in the manuscript.

Author’s response: Because the rarefaction curves are shown, we believe that reporting the annotated reads for each taxonomic level is unnecessary and redundant.

Please, revise all the manuscript for a correct use of the abbreviations. Following some examples:

Line 62: please, explain MT abbreviation

Author’s response: We have followed this suggestion in the revised manuscript such that the full form of abbreviation is used.

Line 68: please, explain SRS abbreviation

Author’s response: We have included the full name of SRS and have avoided abbreviations.

Line 77: DC is already explained before

Author’s response: We have revised this section.

Line 117: DC and MT are already defined

Author’s response: We have revised this section to delete the abbreviation.

Please, revise all the figure legends. They should be self-explanatory.

Author’s response: We have significantly revised the figures and hope this iteration will be appealing and acceptable for publication.

(e.g. Figure 1, please, order properly the samples (DCG1, MTG1, DCG2, MTG2, DCG3, MTG3) and explain name in the legend.)

Author’s response: We have revised the figures to order the samples as suggested.

Reviewer 2 Report

The manuscript presents a modified approach for studying in situ microbial/fungal diversity in U-contaminated soils. The authors describe a method utilizing agar-based microbial traps/diffusion chambers for examining microbial diversity over time. The paper was clearly written with the conclusions succinctly described and mostly supported by the data and results.

There are two aspects of the manuscript that could be improved/clarified:

(1) It would be helpful for the authors to include some pictures, diagrams, images of the MT/DC experimental setup used for the study. It was difficult to visualize the agar and membrane arrangement based on the text description alone. This could be added to the supplementary information file and would greatly improve the ability of other groups to use this approach for similar studies.

(2) Its not overly clear how the results of the statistical analysis were used to assess to relative weight of the difference between treatments. Specifically in reference to Figure 7, the PERMANOVA results included an R-value with an associated p-value that seemed relatively high (~0.067). Within the discussion section, it was not clear whether the authors made a conclusion based on these values (i.e., sig or non-significant differences in alpha or beta indices across groups).

A couple of other minor typos/corrections as follows:

--Line 99: change “to resist” to “to be resistant to” (also in Line 196)

--Line 259: ‘was the predominant of the rest’ is slightly awkward grammatically. Need to rephrase

--Typo, Line 317: 'he' should be 'the'

Author Response

Please see attached file, thanks.

Following is our response to the reviewer’s comments. Shown in black color are the reviewer’s comments and the author’s responses appear in red color.

Reviewer # 2

The manuscript presents a modified approach for studying in situ microbial/fungal diversity in U-contaminated soils. The authors describe a method utilizing agar-based microbial traps/diffusion chambers for examining microbial diversity over time. The paper was clearly written with the conclusions succinctly described and mostly supported by the data and results.

There are two aspects of the manuscript that could be improved/clarified:

(1) It would be helpful for the authors to include some pictures, diagrams, images of the MT/DC experimental setup used for the study. It was difficult to visualize the agar and membrane arrangement based on the text description alone. This could be added to the supplementary information file and would greatly improve the ability of other groups to use this approach for similar studies.

Author’s response: To address the confusion around the experimental plan, we have included a newly generated figure 1 that outlines the entire protocol as a schematic workflow.

(2) Its not overly clear how the results of the statistical analysis were used to assess to relative weight of the difference between treatments. Specifically in reference to Figure 7, the PERMANOVA results included an R-value with an associated p-value that seemed relatively high (~0.067). Within the discussion section, it was not clear whether the authors made a conclusion based on these values (i.e., sig or non-significant differences in alpha or beta indices across groups).

Author’s response: All statistical analysis on the metagenomic sequence data was performed using MicrobiomeAnalyst and the obtained values are representative of this workflow. Our interpretation of data in the discussion is not solely based on one form of analysis but rather on the entirety of the analysis performed, which includes relative abundances over generations of DC and MT, alpha and beta diversity, as well as differential statistical analysis.

A couple of other minor typos/corrections as follows:

--Line 99: change “to resist” to “to be resistant to” (also in Line 196)

Author’s response: We have revised the sentence as suggested.

--Line 259: ‘was the predominant of the rest’ is slightly awkward grammatically. Need to rephrase

Author’s response: We have revised the sentence as suggested.

--Typo, Line 317: 'he' should be 'the'

Author’s response: We have revised the sentence as suggested.

Round 2

Reviewer 1 Report

Authors have improved the manuscript according to my suggestions. Unfortunately, at this stage to be suitable for publication, minor revisions are still needed. 

Line 112: Please, give more details on the soil sampling. What’s the scheme used for sampling? How many samples? Is it a pooled sample?

Line 162: Please correct the sentence, it’s not a “16S metagenomics”.

Authors should motivate in the manuscript the use of QIIME instead of QIIME2. QIIME is an old tool, no longer supported. From January 2018, QIIME2 has succeeded QIIME.

Figures 2 and 6 needs improvements. Following some suggestions:

-Please, revise the legend of figures 2A and B, there is “Others” repeated twice. Define “Other”, what’s included? Authors should define under which relative abundance taxa are not shown and included in “Others”.

-Please, report in the legend only the taxa names, remove “D_1_” and “D_5_” from figure 2 legend and “p_” and “g_” from figure 6 legend.

-Are there unclassified reads only for figure 6B? Please, report the unclassified reads for all the figures.

-Please replace “Class” with “Phylum” in figure 2A and 6A legends, and “Species” in figure 2B and 6B legends. 

As I already suggested in the previous revision:

-Authors should report the number of annotated reads for each taxonomic level they have showed in the manuscript.

Authors suggest to not shown these results because reporting the rarefaction curve. Rarefaction curve shows the number of different species as a function of the number of samples, instead they should report in figures 2 and 6 the total percentage of annotated reads at phylum and genus levels. Probably, at phylum level this percentage should be really close to 100%, but at genus level will be certainly lower. For the same reasons, I previously suggested to shown the percentage of unannotated reads ad both levels.

Author Response

Following is our response to the reviewer’s comments. Shown in black color are the reviewer’s comments and the author’s responses appear in red color.

Reviewer # 2

Line 112: Please, give more details on the soil sampling. What’s the scheme used for sampling? How many samples? Is it a pooled sample?

Author’s response: Details on soil sampling, including historical levels of contamination with heavy metals, such as uranium, has now been added in the revised manuscript.

Line 162: Please correct the sentence, it’s not a “16S metagenomics”.

Author’s response: This has been corrected in the resubmission.

Authors should motivate in the manuscript the use of QIIME instead of QIIME2. QIIME is an old tool, no longer supported. From January 2018, QIIME2 has succeeded QIIME.

Author’s response: We have provided further clarity on the bioinformatic analysis. Note that we’re not strictly using either QIIME or QIIME2. Rather, we’re using some of the tools from QIIME, but also some other tools that we’ve found work better. However, as you state, QIIME is no longer supported since 2018, we have revised this section to indicate QIIME2 was used. Overall, it can be stated that we used an updated sub-OTU pipeline for improved annotations. SILVA is a reference database, independent of the methods used.

-Please, revise the legend of figures 2A and B, there is “Others” repeated twice. Define “Other”, what’s included? Authors should define under which relative abundance taxa are not shown and included in “Others”.

Author’s response: This discrepancy seems to have been introduced from the MicrobiomeAnalyst tool. We have checked our files and the following scheme applies to metagenomic data figures: “Other” has been renamed as “Non-annotated OTUs” and “Others” have been renamed as “Taxa below relative abundance threshold”.

-Please, report in the legend only the taxa names, remove “D_1_” and “D_5_” from figure 2 legend and “p_” and “g_” from figure 6 legend.

Author’s response: These figures have been revised.

-Are there unclassified reads only for figure 6B? Please, report the unclassified reads for all the figures.

Author’s response: The unclassified reads are labeled as “Non-annotated OTUs” in all the metagenomics data figures.

-Please replace “Class” with “Phylum” in figure 2A and 6A legends, and “Species” in figure 2B and 6B legends.

Author’s response: This has been revised.

As I already suggested in the previous revision:

-Authors should report the number of annotated reads for each taxonomic level they have showed in the manuscript. Authors suggest to not shown these results because reporting the rarefaction curve. Rarefaction curve shows the number of different species as a function of the number of samples, instead they should report in figures 2 and 6 the total percentage of annotated reads at phylum and genus levels. Probably, at phylum level this percentage should be really close to 100%, but at genus level will be certainly lower. For the same reasons, I previously suggested to shown the percentage of unannotated reads ad both levels.

Author’s response: As suggested by the reviewer, this information has now been added in table 1. This shows total read counts for both bacterial and fungal domains and counts that were taxonomically binned at the phylum and genera levels across different generations of the DC and MT, respectively.